# Minimally Invasive Laminate Veneer Therapy for Maxillary Central Incisors

**DOI:** 10.3390/medicina59030603

**Published:** 2023-03-18

**Authors:** Gerardo Guzman-Perez, Carlos A. Jurado, Francisco Azpiazu-Flores, Kelvin I. Afrashtehfar, Akimasa Tsujimoto

**Affiliations:** 1CEMRO Periodontics Department, Morelia 58880, Mexico; 2Department of Prosthodontics, University of Iowa College of Dentistry, Iowa City, IA 52242, USA; 3Department of Restorative Dentistry, University of Manitoba Dr. Gerald Niznick College of Dentistry, Winnipeg, MB R3E 0W2, Canada; 4Division of Restorative Dental Sciences, Clinical Sciences Department, Ajman College of Dentistry, Ajman City P.O. Box 346, United Arab Emirates; 5Department of Reconstructive Dentistry and Gerodontology, University of Bern School of Dental Medicine, 3010 Bern, Switzerland; 6Department of Operative Dentistry, University of Iowa College of Dentistry, Iowa City, IA 52242, USA; 7Department of General Dentistry, Creighton University School of Dentistry, Omaha, NE 68102, USA

**Keywords:** esthetic dentistry, dental ceramics, dental veneers, minimally invasive dentistry, restorative dentistry

## Abstract

Minimally invasive dentistry is a considered process that requires the clinician to be prepared with the ideal sequence and the tools needed. This report describes a well-planned ultraconservative approach using only two ceramic laminate veneers for the maxillary central incisors to significantly improve the patient’s overall smile. A 30-year-old female presented with the chief complaints of having diastemas between the central and lateral incisors as well as incisal wear. Diagnostic wax-up and mock-up were performed, and the patient approved the minimally invasive treatment with veneers only for central incisors. A reduction guide aided the conservative tooth preparations, and hand-crafted feldspathic veneers were bonded under total isolation with a rubber dam. The two final conservative veneers significantly improved the smile and fulfilled the patient’s expectations. Following proper planning and sequencing, predictable outcomes were obtained and fulfilled the patient’s esthetic demands. Minimally invasive restorative dentistry with only two single veneers can impact the entire smile frame. Overtreatment in the esthetic zone is unnecessary to meet a patient’s esthetic expectations.

## 1. Introduction

Minimally invasive dentistry is a popular approach that emphasizes the conservative use of dental techniques and materials to preserve natural tooth structure and maintain the integrity of dentition [1]. Restorative dentistry can benefit from this approach, with composite resins offering advantages such as tooth structure preservation and avoidance of tooth grinding to generate a path of insertion and obtaining a minimum thickness required for an indirect restorative material. However, ceramics might be preferred for their superior longevity, esthetics, and ability to restore the already missing tooth structure [2]. The choice between composite and ceramic materials should be made on a case-by-case basis, considering the extent of tooth damage, esthetic goals, and patient preferences. A thorough evaluation by a dental professional is necessary to determine the most appropriate material for each individual case [2,3]. Novel technologies and materials have made it possible to take a more conservative approach to dental treatment. Adhesive bonding agents, composite resins, and ceramic restorations can help reduce the need for more invasive procedures such as crown placement or extraction [3]. Ultrathin ceramic laminate veneers have become a minimally invasive solution to improve the esthetics of anterior teeth without significant preparation, allowing for bonding purely in the enamel structure and preservation of the natural tooth structure [4]. Overall, minimally invasive dentistry can provide conservative and esthetic dental treatment. The selection between composite and ceramic materials should be made based on individual factors, and ultrathin ceramic veneers can be a minimally invasive solution for anterior teeth esthetics.

Dental esthetic treatment is a rapidly growing field, with patients increasingly seeking procedures to enhance their smile’s appearance. Maxillary teeth size, shape, and spacing are essential in dental and facial esthetics [5]. Among these teeth, the maxillary central incisors are particularly important. They are often the focus of attention when achieving ideal position, size, proportion, and symmetry, given their critical role in the smile and smile arc [6]. Several studies have shown that the position of the incisal edges of the central incisors plays a significant role in creating a youthful, attractive smile [7]. Moreover, black triangles and spaces between central incisors are often associated with the aging population, and can negatively impact the overall aesthetics of the face and smile complex [8]. The relationship between the gingival margin, the central incisors, and the maxillary lip line is crucial to creating an attractive smile in patients. For instance, it has been established that the upper lip should rest on the gingival margin of the maxillary central incisors to achieve an attractive smile in females [9]. Therefore, it is imperative that dental practitioners prioritize the treatment of central incisors when restoring or enhancing a patient’s smile. Thus, to achieve optimal results in dental esthetic treatment, clinicians must carefully consider the patient’s facial features, tooth size, and shape, as well as their overall oral health. Using minimally invasive techniques, such as laminate veneer therapy, is a promising approach to achieving excellent esthetic outcomes while preserving as much healthy tooth structure as possible [4,10].

Tooth preparation for ceramic laminate veneer restorations can be complex, and adequate planning is essential to avoid potential complications. Accurate tooth reduction guides can help achieve a uniform reduction and prevent overpreparation, which can result in undesirable outcomes [10]. These guides provide a well-controlled removal of tooth structure, which is crucial for the success of the restoration and preservation of the natural tooth structure [11]. Tooth reduction guides can be created using different materials, such as PVS, acrylic resin, thermoplastic sheets, or cast metal. They allow evaluation of tooth reduction from all aspects, including incisal, facial, and interproximal, which helps to achieve an optimal restoration outcome [12]. Moreover, dental practitioners have numerous ceramic options for the fabrication of ceramic laminate veneers, including feldspathic porcelain, leucite, lithium disilicate, zirconia, and combinations. These materials provide different mechanical properties, each with indications for use [13]. Studies have shown that ceramic laminate veneer restorations bonded to enamel tissue have a high survival rate. A retrospective study that evaluated the long-term survival of feldspathic veneers found a 96% survival rate at 16-year follow-up, indicating that this treatment modality and material are considered predictable [14,15]. Therefore, dental practitioners must understand the different aspects of tooth preparation and the available ceramic materials to ensure predictable optimal patient outcomes.

This article aims to present a case report of an ultraconservative dental treatment that involves the placement of only two feldspathic veneers for maxillary central incisors. The protocol, in this case, followed a well-planned sequence involving a diagnostic mock-up, minimal tooth reduction guided by a reduction guide, the fabrication of highly esthetic hand-crafted veneers, and bonding with rubber dam isolation. The results demonstrate that this ultraconservative approach can significantly improve smile esthetics while minimizing damage to the natural tooth structure. This treatment option can be an excellent alternative to more invasive procedures and can lead to high patient satisfaction.

## 2. Materials and Methods

A 30-year-old female patient presented to the clinic with the chief complaint of “I want to improve my smile and close the space in between my teeth” (Figure 1). 

After evaluation, the patient was diagnosed with incisal wear for both maxillary central incisors as well as diastemas between the central incisors and between the central and lateral incisors. The patient was questioned if there was a reason for only having central incisors with wear and she claimed to bite her nails whenever she is stressed out. The patient was informed to stop biting her nails otherwise any restoration can fail, and she committed to stopping that parafunctional habit. The patient was initially offered an orthodontic evaluation to investigate the possibility of closing the spaces and then a reevaluation for restorative treatment; however, the patient disliked the idea of having orthodontic appliances in her mouth and requested restorative therapy alone. Thus, an orthodontic solution was out of the question. 

The patient was informed of the need for a diagnostic mock-up to assess restorative options. The mock-up was to evaluate if only two (i.e., both central incisors) or four (i.e., central and lateral incisors) teeth required laminate veneers. Diagnostic impressions were made with PVS material (Virtual, Ivoclar Vivadent, Schaan, Liechtenstein) and poured out twice with type IV stone (Fujirock, GC America), followed by facebow record, and mounted in a semi-adjustable articulator (Artex CR, Amann, Girrbach). Two diagnostic wax-ups (Gray Wax GEO Classic, Renfert) were performed; the first was for both maxillary central incisors, and the second with both maxillary central and lateral incisors. Diagnostic mock-ups were provided in the mouth with a putty guide (Elite P&P, Zhermack), and the patient evaluated the results. The patient was pleased with the reconstructive results for the restoration of only the central incisors and asked to begin the treatment (Figure 2).

The putty reduction guide (Hydrorise putty, Zhermack, Badia Polesine, Italy) was made based on the diagnostic wax-up for both central incisors. Ultraconservative tooth preparations were provided with the diagnostic mock-up on, starting with horizontal reduction grooves. The reduction guide was intermittently placed to evaluate the amount of tooth removal. Cord 0 (Ultrapak, Ultradent Products Inc, South Jordan, UT, USA) was packed prior to finalizing the cervical third of the preparation, and the final tooth preparations were polished using discs (Sof-Lex Discs, 3M Oral Care, St Paul, MN, USA) (Figure 3). 

The final impression of the tooth abutments used light- and heavy-body consistency PVS materials (Virtual 380, Ivoclar Vivadent, Schaan, Liechtenstein) (Figure 4).

The final impression was poured out, and the master cast was fabricated with type IV stone (Fujirock, GC America). The hand-crafted porcelain feldspathic laminate veneers (Noritake Super Porcelain EX-3, Kuraray Noritake Dental Inc., New York, NY, USA) were manufactured following the diagnostic wax-up dimensions (Figure 5). 

When creating ceramic dental restorations, color acquisition is critical in achieving a natural-looking result, particularly for anterior maxillary teeth that should be polychromatic. Color acquisition involves using shade guides, spectrophotometers, and other devices to determine the shade and hue of the natural tooth and the translucency and opacity required for the restoration to blend seamlessly with adjacent teeth. Factors affecting color acquisition include lighting conditions, background color, and the clinician’s ability to perceive and communicate color information accurately. Dental laboratories may also use specialized computer programs and digital imaging technologies to aid in color matching and customization of the restoration. Overall, color acquisition is a complex process that requires skill and precision to achieve optimal aesthetic outcomes. Nevertheless, in this case, the teeth’s natural color was bright, eliminating the need for any significant color changes or compensation during restoration. 

The ceramic veneers were tried on, the patient approved the esthetics, and she asked to proceed with the permanent cementation process. The intaglio surface of the laminate veneers was treated first with hydrofluoric acid (Porcelain Etch, Ultradent Products, Inc, South Jordan, UT, USA) for 60 s, followed by rinsing and air-drying. Then, the restorations were cleaned with phosphoric acid (Total Etch, Ivoclar Vivadent, Schaan, Liechtenstein) and silanized (Monobond Plus, Ivoclar Vivadent, Schaan, Liechtenstein) for 60 s (Figure 6). 

Total isolation was provided with a rubber dam (Dental Dam, Nic Tone) covering the maxillary anterior region and retained with claps (Clamp 2, Hu-Friedy). The central incisors also received clamps (Clamp 212, Hu-Friedy) along the gingival contours, and the teeth were treated first with 37% phosphoric acid (Total Etch, Ivoclar Vivadent, Schaan, Liechtenstein) for 15 s and gently air dried, followed by primer application (Primer Optibond FL, Kerr, Orange, CA, USA), and excess was gently removed. 

Then, adhesive (Adhesive Optibond, FL Kerr, Orange, CA, USA) was applied, and the ceramic restorations were cemented with light shade resin cement (Variolink Esthetic LC, Ivoclar Vivadent, Schaan, Liechtenstein). Excess cement was removed, and each restoration received an initial tack cure within 5 s. Additional excess cement was removed with floss, and the final light-cure was provided with 20 s on the facial surface, 20 s on the mesial, and 20 s on the distal surface (Figure 7). 

The patient was pleased with the feldspathic veneer restorations’ shape, shade, and contours (Figure 8). The patient was provided with an occlusal night guard to protect the restorations from potential chipping or delamination.

## 3. Discussion

Minimally invasive treatment with composite resins has gained popularity in recent years due to its ability to preserve natural tooth structure and achieve satisfactory esthetic and functional outcomes. Composite resin restorations avoid tooth reduction and can address various cosmetic and functional issues. Ceramic laminate veneers are thin, custom-made shells that require tooth reduction and are bonded to the buccal surface of the teeth to improve their appearance. They are indicated in cases where tooth structure is already missing or when significant cosmetic enhancements are required. The decision between minimally invasive treatment with composite resins or ceramic veneers should be made on a case-by-case basis, considering factors such as the extent of the compromised tooth structure, esthetic goals, and patient preferences. Composite resin restorations are preferred when the natural tooth structure needs to be preserved, while ceramic laminate veneers are indicated for more severe cases or when significant cosmetic enhancements are needed [16]. Overall, the choice between composite and ceramic veneers should involve a thorough evaluation by a dental professional to determine the most appropriate treatment plan while minimizing tooth reduction and preserving as much natural tooth structure as possible. A comprehensive evaluation is required when ceramic veneers might be a valid option, including photographs, diagnostic wax-up, and mock-up, and can provide valuable information on the amount of tooth reduction required before irreversible treatment is performed [16]. The diagnostic wax-up is the first step, which can be performed using the additive technique, creating the possibility of a diagnostic mock-up. However, this may not be possible in situations where the tooth is rotated or tilted or there are interproximal gaps between the teeth [17]. 

Fortunately, in the presented case, the patient only had two abraded maxillary central incisors with a combined diastema closure between the distal surface of the central and the mesial surface of the lateral teeth using 0.3–0.5 mm restorative veneers. In other words, the gap between the central and lateral teeth was relatively small, less than 2 mm. Therefore, a combined gap closure with veneers was a suitable and conservative option. If the gap was more extensive, restoring both lateral incisors may have been necessary to maintain a harmonious size between the teeth. Additionally, the restorations required for the central incisors were relatively small, so regular porcelain could be used. If the restorations required more extended sizes, it might have been necessary to use ceramics with higher flexural strength. Additionally, the teeth’s natural color was bright, eliminating the need for any significant color changes or compensation during restoration. The dental arch was even, with minimal tilting or rotation of the teeth. This allowed for a more predictable and conservative approach to the restoration. Finally, the patient had a medium smile line, which allowed for an equigingival finish line, further reducing the procedure’s invasiveness. Hence, the additive wax-up and intraoral mock-up were feasible.

A diagnostic mock-up allows patients to see and feel the proposed restorations in their mouths before undergoing any irreversible therapy, providing them with a real test of how the final result will look and feel. Patients can request modifications before the tooth preparation stage [18]. In our case, the patient was satisfied with the diagnostic mock-up for the central incisors and agreed to move forward with the tooth preparations. Using reduction guides in tooth preparation allowed us to achieve a uniform and precise reduction, avoiding overpreparation of teeth, which may lead to undesirable outcomes [10]. The final restorations fulfilled the patient’s expectations, as she achieved the desired esthetic result with minimal intervention, enhancing her overall smile.

Tooth reduction guides are crucial in ensuring an ultraconservative tooth preparation that maintains the tooth structure and minimizes the risk of overpreparation. They also help to achieve accurate and consistent reduction, especially for thin laminate veneer restorations [19]. Inexperienced clinicians may find the use of reduction guides technically more straightforward and more predictable than freehand tooth preparation. It is important for clinicians to be familiar with different types of reduction guides and their fabrication methods. Constant measurement with a periodontal probe can also ensure controlled and precise reduction during the tooth preparation process. In this case, the guide used evaluated the facial reduction from an incisal view. It helped to smoothen the tooth surface without removing the incisal length of the tooth, which was unnecessary due to the incisal wear present in the patient’s teeth.

Among different types of ceramic laminate veneers, hand-crafted feldspathic porcelain veneers have shown excellent long-term survival rates, with a 95.5% cumulative survival rate after ten years [20]. Hand-crafted porcelain veneers have been recommended for a range of cosmetic dental issues, such as tooth discoloration, mild incisal wear, and interproximal spacing, due to their highly esthetic and functional results [21]. The patient, in this case, was treated with hand-made porcelain veneers that matched her adjacent teeth, providing a natural look and improving her overall smile without requiring additional tooth treatments. Using these veneers with minimal tooth reduction facilitated a minimally invasive approach to achieve the patient’s desired outcome.

Finally, it is essential to note that the success of any dental treatment depends not only on the technical aspects but also on the patient’s satisfaction [22,23,24,25,26,27]. Therefore, thoroughly discussing the patient’s expectations and desires before initiating any treatment is crucial to ensure the desired outcome [28,29,30,31,32]. In this case, the patient was highly satisfied with the final outcome and reported a significant improvement in her self-confidence and overall satisfaction with her smile. This confirms the importance of patient-centered care in achieving successful clinical outcomes.

Overall, this report demonstrates the efficacy of an ultraconservative approach to esthetic dental treatment using only two feldspathic veneers for maxillary central incisors. The treatment sequence, including diagnostic mock-up, minimal tooth reduction with guides, hand-crafted veneer fabrication, and bonding protocol using rubber dam isolation, can provide highly esthetic and functional results while minimizing the amount of tooth structure removed. The long-term survival rates of porcelain laminate veneers and the use of rubber dam isolation and selective etching techniques further support the reliability and predictability of this treatment modality.

### 3.1. Recommendations

Incorporating patient-centered care and communication is crucial to meeting the patient’s desires and expectations. Clinicians should also evaluate patients’ periodontal health and occlusion before initiating restorative treatment. Additionally, clinicians should consider using reduction guides to achieve precise and uniform tooth reduction during tooth preparation. This can minimize the amount of tooth structure removed while still achieving the desired esthetic outcome.

Clinicians should also stay up to date with the latest advancements in materials and technologies for minimally invasive restorative dentistry. By doing so, they can provide their patients with the most effective and efficient treatment options available. Finally, clinicians need to discuss the cost of treatment with patients to ensure that they can make informed decisions regarding their dental care. By following these recommendations, clinicians can provide highly esthetic and functional results while minimizing the amount of tooth structure removed.

### 3.2. Future Research

To further advance our understanding of minimally invasive restorative dentistry, larger-scale studies are needed to investigate these techniques’ long-term durability and success. Research should also focus on the impact of minimally invasive restorative dentistry on the periodontal health of patients and the occlusal function and long-term stability of restorations. Additionally, studies should examine the cost-effectiveness of minimally invasive restorative dentistry compared to traditional restorative techniques.

More specifically, research should investigate using newer technologies, such as digital impressions, computer-aided design, and computer-aided manufacturing (CAD/CAM) systems, in minimally invasive restorative dentistry. Studies should also investigate using alternative materials, such as zirconia and lithium disilicate, for veneer fabrication. These materials may offer improved mechanical and esthetic properties, further enhancing the success and longevity of minimally invasive restorative treatments.

## 4. Conclusions

Minimally invasive dentistry is an essential aspect of modern dental practice, and this clinical report highlights the benefits of this approach for improving patient satisfaction and outcomes aided by adhesive dentistry. By using only the minimum number of necessary restorations and preserving tooth structure as much as possible, clinicians can significantly improve the overall smile without overtreatment. In this case, using two single laminate veneers for maxillary central incisors demonstrated how even small changes can enhance the harmony between face and smile, significantly boosting self-confidence and overall satisfaction.

This case report also highlights the importance of using proven techniques and materials to achieve predictable outcomes. Using a diagnostic wax-up, ultraconservative tooth preparations, hand-crafted porcelain laminate veneers, and luting cementation under rubber dam isolation all contributed to a successful and esthetically pleasing result. Future research in this area could focus on developing even more minimally invasive techniques and materials that can provide even better long-term outcomes while preserving as much tooth structure as possible. By continuing to prioritize minimally invasive dentistry, clinicians can provide their patients with the highest quality care while minimizing the risk of complications and overtreatment.

## Figures and Tables

**Figure 1 medicina-59-00603-f001:**
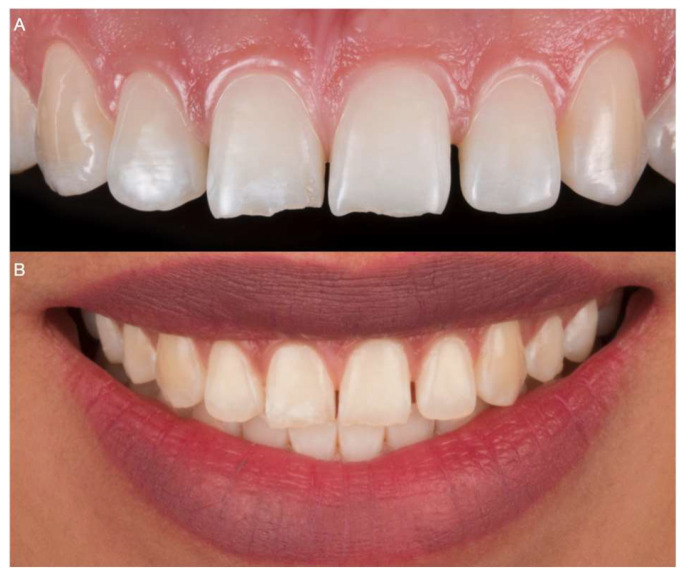
Frontal view of the initial situation. (**A**) Maxillary intraoral aspect. (**B**) Extraoral smile.

**Figure 2 medicina-59-00603-f002:**
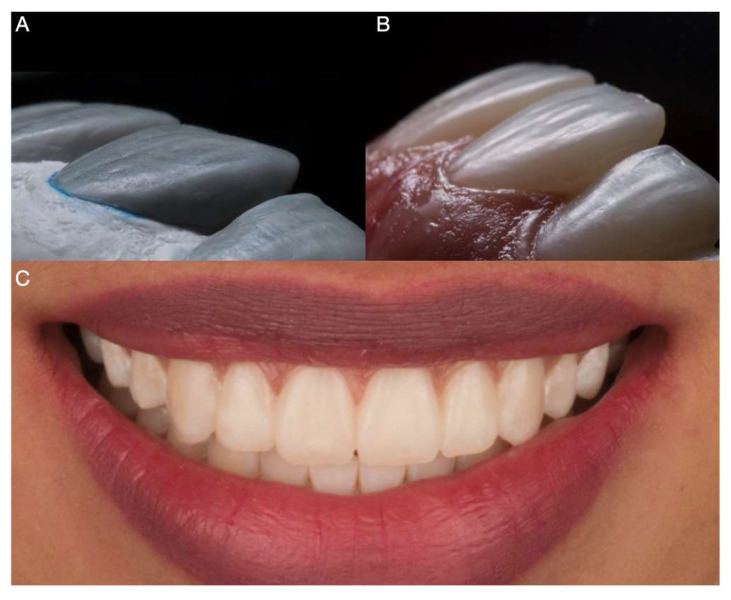
Diagnostic mock-up. (**A**) Diagnostic wax-up. (**B**) Intraoral close-up of diagnostic mock-up. (**C**) Extraoral view of diagnostic mock-up while smiling.

**Figure 3 medicina-59-00603-f003:**
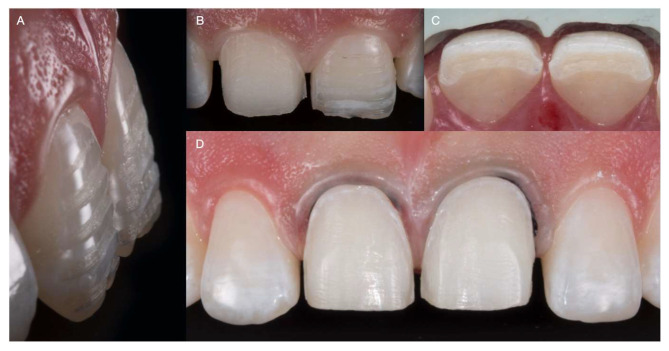
Ultraconservative tooth preparations. (**A**) Lateral view of horizontal reduction grooves. (**B**) Tooth preparations in process. (**C**) Incisal view of preparation evaluated with a reduction guide. (**D**) Finalized tooth preparations (gingival retraction cords in both centrals).

**Figure 4 medicina-59-00603-f004:**
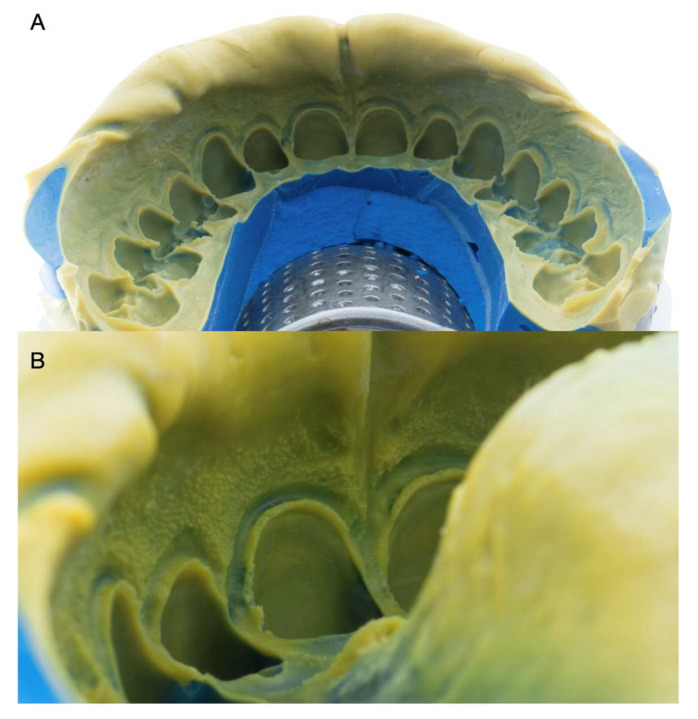
Final impression. (**A**) Maxillary arch impression. (**B**) Impression close-up.

**Figure 5 medicina-59-00603-f005:**
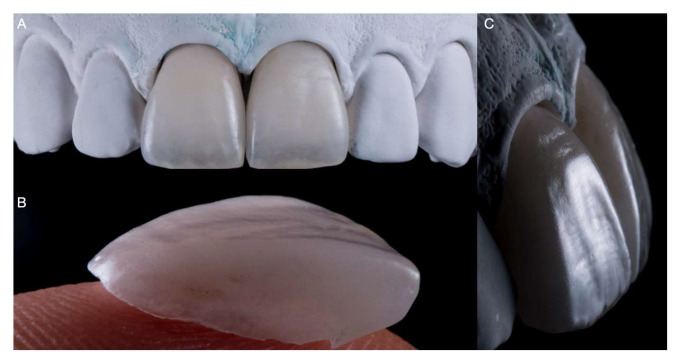
Permanent restorations: ultrathin ceramic laminate veneers. (**A**) Frontal view of veneers on master cast. (**B**) Close-up of a single veneer. (**C**) Lateral view of veneers on master cast.

**Figure 6 medicina-59-00603-f006:**
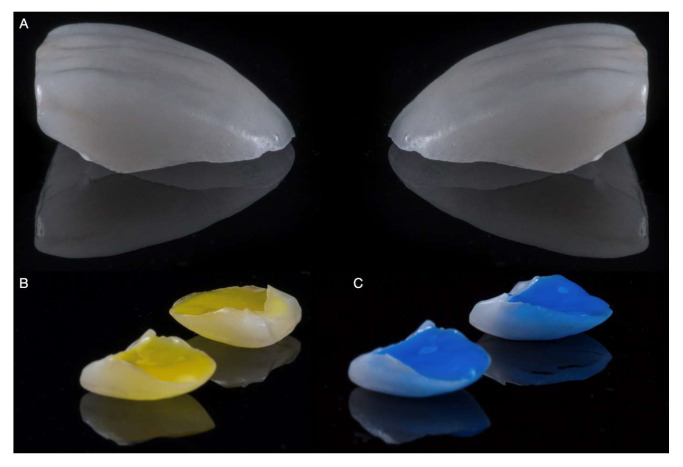
Intaglio surface ceramics treatment. (**A**) Indirect restorations prior to surface treatment. (**B**) Hydrofluoric acid treatment. (**C**) Phosphoric acid treatment.

**Figure 7 medicina-59-00603-f007:**
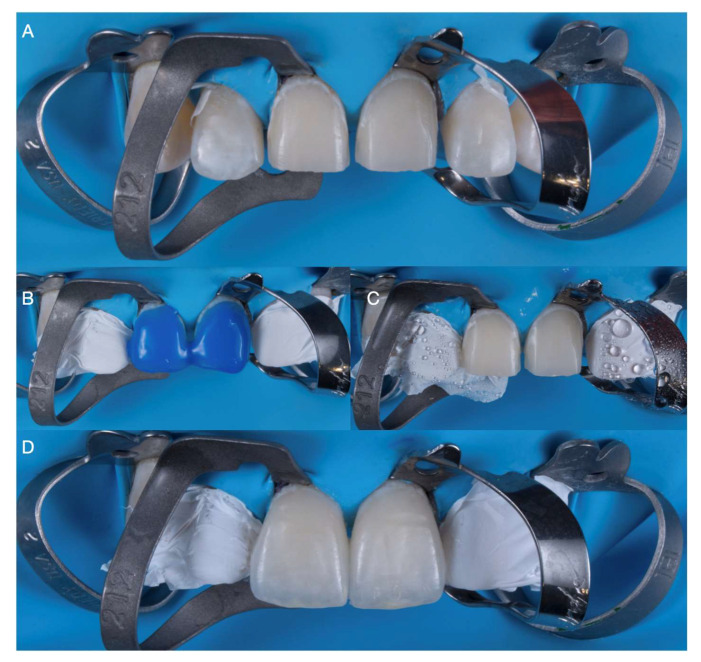
Bonding–luting process. (**A**) Rubber dam isolation. (**B**) Phosphoric acid treatment of prepared tooth surface. (**C**) Rinsed. (**D**) Final cementation of veneers.

**Figure 8 medicina-59-00603-f008:**
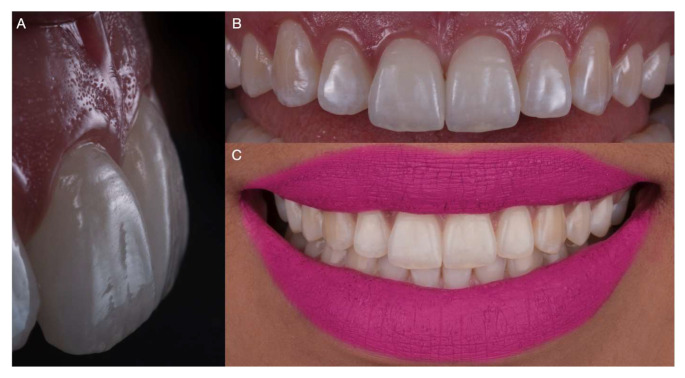
Clinical appearance of the installed bonded laminate veneers. (**A**) Lateral view. (**B**) Maxillary intraoral aspect. (**C**) Extraoral final smile.

## Data Availability

Not applicable.

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
