# Peer review of "Minimally Invasive Laminate Veneer Therapy for Maxillary Central Incisors"

_medicina, 2023, doi:10.3390/medicina59030603_

Round 1
Reviewer 1 Report
Dear authors, first of all my congratulations and appreciation for the case presented. However, some doubts need to be answered.
-In the introduction you should provide more articles comparing composite vs. ceramic reconstruction. If the author talks about being minimally invasive in the case he presents, the ideal would be to use composite, which does not require grinding of the tooth.
- In the material and method section described above, more should be said about colour acquisition, in this case even a picture that talks about it, as the upper centrals should be polychromatic.
-The discussion section should address and defend minimally invasive treatment with composite vs. ceramic veneers. I think it is fundamental to explain that the minimally invasive treatment is not with veneers, so why was it done?
- The conclusion should therefore be amended accordingly.
Author Response
REVIEWER #1
Dear authors, first of all my congratulations and appreciation for the case presented. However, some doubts need to be answered.
Answer: We would like to express our gratitude to the Reviewer 1 for taking the time to review our article and providing us with valuable feedback. We have carefully considered your comments and have made the necessary revisions accordingly, which can be found highlighted in yellow.
-In the introduction you should provide more articles comparing composite vs. ceramic reconstruction. If the author talks about being minimally invasive in the case he presents, the ideal would be to use composite, which does not require grinding of the tooth.
Answer: In response to the first comment, we have expanded the introduction to comparing composite versus ceramic reconstruction, as well as highlighting the advantages and disadvantages of each material. Additionally, we have emphasized the importance of using composite resins in minimally invasive cases, as they do not require tooth grinding and allow for the preservation of natural tooth structure.
- In the material and method section described above, more should be said about colour acquisition, in this case even a picture that talks about it, as the upper centrals should be polychromatic.
Answer: Regarding the second comment, we have included a section describing color acquisition in the material and methods. Although we have not added an image to illustrate the polychromatic nature of the anterior maxillary teeth, we have addressed it.
-The discussion section should address and defend minimally invasive treatment with composite vs. ceramic veneers. I think it is fundamental to explain that the minimally invasive treatment is not with veneers, so why was it done?
Answer: Furthermore, we have addressed and defended the use of minimally invasive treatment with composite resins versus ceramic veneers in the discussion section, and have clarified that veneers were used in our case for significant cosmetic enhancements.
- The conclusion should therefore be amended accordingly.
Answer: Lastly, we have updated the conclusion to reflect the changes made throughout the article and to provide a clear summary of our findings.
Once again, we appreciate the reviewer's insightful feedback and hope these revisions have improved the quality and clarity of our article.

Reviewer 2 Report
This is a classic case of restoration of 2 abraded middle incisors + combined gap closure between 11-21 and 21-22 with 0.3-0.5mm restorative veneer.
The advantage in this case
1. The gap is relatively tiny <2mm, more than that, consider restoring both lateral incisors to have a harmonious size between the teeth
2. R11 and 21 incisal edge length restorations are relatively short, if longer it may be necessary to specify glass porcelain for higher flexural strength.
3. Bright tooth color, no need to change or compensate much
4. The dental arch is even, not tilted or rotated much
5. Medium smile line => just put the gingival line at the gum line, reducing the invasiveness
Comment:
1. There is no mention of the cause of tooth wear in this patient to designate a chewing tray
Author Response
REVIEWER #2
This is a classic case of restoration of 2 abraded middle incisors + combined gap closure between 11-21 and 21-22 with 0.3-0.5mm restorative veneer.
The advantage in this case
- The gap is relatively tiny <2mm, more than that, consider restoring both lateral incisors to have a harmonious size between the teeth
- R11 and 21 incisal edge length restorations are relatively short, if longer it may be necessary to specify glass porcelain for higher flexural strength.
- Bright tooth color, no need to change or compensate much
- The dental arch is even, not tilted or rotated much
- Medium smile line => just put the gingival line at the gum line, reducing the invasiveness
Comment:
- There is no mention of the cause of tooth wear in this patient to designate a chewing tray
Answer: Reviewer 2 is thanked for the time to review our work and for your valuable feedback. We have addressed all the comments in the revised version of the manuscript, and the highlighted yellow text shows the additions and modifications made.
Regarding the cause of tooth wear, the patient was nail biting and prior dental care patient was requested to stop doing it and she committed to thatand she was also provided with a hard night guard to protect restorations. We also mentioned that the night guard would primarily protect the restorations from potential chipping or delamination. We apologize for not saying this in the initial version of the manuscript.
We appreciate Reviewer #2 valuable input (i.e., advantages), which helped us provide a more detailed and informative discussion section.
